

# Diversity analysis of gut microbiota in osteoporosis and osteopenia patients

Jihan Wang[1,*], Yangyang Wang[2,*], Wenjie Gao[1,*], Biao Wang[1], Heping Zhao[1], Yuhong Zeng[1], Yanhong Ji[3] and Dingjun Hao[1]

[1] Hong Hui Hospital, Xi'an Jiaotong University, Xi'an, China
[2] The Tenth Research Institute of Telecommunications Technology, Xi'an, China
[3] Department of Pathogenic Biology and Immunology, Key Laboratory of Environment and Genes Related to Diseases, Ministry of Education of China, School of Basic Medical Sciences, Xi'an Jiaotong University Health Science Center, Xi'an, China
[*] These authors contributed equally to this work.

Corresponding authors
Yanhong Ji, jiyanhong@xjtu.edu.cn
Dingjun Hao, haodingjun@126.com

## ABSTRACT

Some evidence suggests that bone health can be regulated by gut microbiota. To better understand this, we performed 16S ribosomal RNA sequencing to analyze the intestinal microbial diversity in primary osteoporosis (OP) patients, osteopenia (ON) patients and normal controls (NC). We observed an inverse correlation between the number of bacterial taxa and the value of bone mineral density. The diversity estimators in the OP and ON groups were increased compared with those in the NC group. Beta diversity analyses based on hierarchical clustering and principal coordinate analysis (PCoA) could discriminate the NC samples from OP and ON samples. Firmicutes, Bacteroidetes, Proteobacteria and Actinobacteria constituted the four dominant phyla in all samples. Proportion of Firmicutes was significantly higher and Bacteroidetes was significantly lower in OP samples than that in NC samples ($p < 0.05$), Gemmatimonadetes and Chloroflexi were significantly different between OP and NC group as well as between ON and NC group ($p < 0.01$). A total of 21 genera with proportions above 1% were detected and Bacteroides accounted for the largest proportion in all samples. The Blautia, Parabacteroides and Ruminococcaceae genera differed significantly between the OP and NC group ($p < 0.05$). Linear discriminant analysis (LDA) results showed one phylum community and seven phylum communities were enriched in ON and OP, respectively. Thirty-five genus communities, five genus communities and two genus communities were enriched in OP, ON and NC, respectively. The results of this study indicate that gut microbiota may be a critical factor in osteoporosis development, which can further help us search for novel biomarkers of gut microbiota in OP and understand the interaction between gut microbiota and bone health.

## INTRODUCTION

Osteoporosis is a type of bone-thinning disorder, characterized by a reduction of bone mass, microarchitecture deterioration and an increased risk of fragility fractures. It is the most common reason for a broken bone among the elderly. As the population grows

and ages, the number of patients with osteoporosis is expected to increase. A decline in bone mineral density (BMD) is the primary cause of fragility fracture (*Lu et al., 2016*). As a metabolic procedure, bone homeostasis relies on a balance between bone formation (osteoblast-regulated) and bone resorption (osteoclast-regulated) (*Chung et al., 2014*; *Harada & Rodan, 2003*). Hereditary characteristics and environmental factors can regulate the complex process of bone metabolism and significantly contribute to age-related bone loss (*Pollitzer & Anderson, 1989*).

Recently, the gut microbiota have attracted attention in connection with metabolic diseases. The human gastrointestinal tract are colonized by rich and dynamic communities of microbes. The microbes has been considered as a critical factor for metabolic disorders including obesity, diabetes, and osteoporosis (*Ejtahed et al., 2016*). Therefore, it may represent a novel potential biomarker for the diagnosis and treatment of metabolic disorders (*Steves et al., 2016*). So far, the effect of gut microbiota on bone health is a relatively new field of research. Several studies have reported it as a regulator of bone mass (*McCabe, Britton & Parameswaran, 2015*; *Sjogren et al., 2012*; *Weaver, 2015*) through mediation of the immune system (e.g., osteoclastogenesis), intestinal calcium absorption and the release of neurotransmitters (e.g., serotonin). A better understanding of structure and function changes of microbes will help us search for novel biomarkers and understand the interaction between gut microbiota and bone mass disorder. However, to our knowledge, it remains unclear how gut microbiota changes in osteoporosis patients.

Traditional methods for research on bacterial community inhabitants include isolation, cultivation, and optical microscopy. These approaches are insufficient to obtain relatively full-scale and accurate results about the structure and diversity of microbiota communities in specific samples because the vast majority of bacteria in fecal samples are anaerobic and cannot be isolated in the laboratory (*Perry et al., 2010*). High-throughput sequencing has recently been used for bacterial diversity analysis (*Li et al., 2016a*; *Li et al., 2016b*). This approach overcomes the limitations of traditional technology and can effectively capture the genomic information of uncultured microorganisms, which may be pathogenic or important for biological processes.

The present study was to explore the bacterial community structure and diversity changes of gut microbiota in patients with primary osteoporosis and primary osteopenia based on 16S rRNA gene sequencing. Results of our research will lay a foundation for searching novel microbe biomarkers and understanding the potential mechanisms of effects of gut microbiota on bone health.

## METHODS

### Subject recruitment and bone mineral density detection

Participants in this study were recruited from Hong Hui Hospital, Xi'an Jiaotong University, Xi'an, China. Dual X-ray absorptiometry (DXA) was performed to detect the bone mineral density of lumbar vertebrae of subjects. We further excluded all patients with any malignancy, chronic liver disease, heart disease, kidney disease, or diabetes. Finally, a total of 18 subjects including six with primary osteoporosis (OP), six with primary

**Table 1  Clinicopathological information of the study participants.**

| Group | Case | Gender | Age | BMD $L_{1-4}$ (g/cm$^2$) | $Z$-score $L_{1-4}$ | $T$-score $L_{1-4}$ |
|---|---|---|---|---|---|---|
| Normal control (NC) | 6 | Female: 5 Male: 1 | 64.80 $\pm$ 5.93 | 0.81 $\pm$ 0.08 | 0.12 $\pm$ 0.45 | -0.42 $\pm$ 0.26 |
| Osteopenia (ON) | 6 | Female: 5 Male: 1 | 67.17 $\pm$ 8.30 | 0.75 $\pm$ 0.04* | $-0.22 \pm 0.50$ | $-2.15 \pm 0.34$* |
| Osteoporosis (OP) | 6 | Female: 5 Male: 1 | 70.00 $\pm$ 7.77 | 0.61 $\pm$ 0.06***## | $-1.18 \pm 0.73$**# | $-3.57 \pm 0.46$***## |

Notes.

Compares with NC group: *$P < 0.05$, **$P < 0.01$. Compares with ON group: #$P < 0.05$, ##$P < 0.01$.

BMD, $Z$-score and $T$-score were collected from dual X-ray absorptiometry detection, $L_{1-4}$ represents lumbar vertebrae 1-4.

BMD, bone mineral density; $Z$-score, the $Z$-score is the comparison to the age-matched normal; $T$-score, the $T$-score is the relevant measure when screening for osteoporosis.

The criteria of the World Health Organization are: Normal is a $T$-score of $-1.0$ or higher; Osteopenia is defined as between $-1.0$ and $-2.5$; Osteoporosis is defined as $-2.5$ or lower.

osteopenia (ON), and six normal controls (NC; as determined by physical examination) were selected for further research (Table 1). None of the 18 participants ingested yogurt, prebiotics, or probiotics during the fecal collection period, nor had they used medication (e.g., antibiotics) within one month of sample collection. The study was approved by Hong Hui Hospital, Xi'an Jiaotong University, Biomedical research ethics committee. Each participant provided his or her written informed consent.

## Fecal sample collection and DNA extraction

Fresh stool samples were collected in sterile boxes, then frozen and stored at $-80\ °C$ for further use. The microbial genome was extracted using QIAamp Fast DNA Stool Mini Kit (Qiagen, Hilden, Germany) according to the manufacturer's instructions. Sample DNA purity and concentration were tested using a Nanodrop 2000 Spectrophotometer.

## 16S rRNA PCR and Illumina sequencing

We amplified the bacterial 16S ribosomal RNA gene V3-V4 region using the TransGen AP221-02 Kit (TransGen, Beijing, China). The following PCR primers were used: 338F 5′-ACTCCTACGGGAGGCAGCAG-3′ and 806R 5′-GGACTACHVGGGTWTCTAAT-3′. The reaction volume (20 μl) comprised 5 × FastPfu Buffer (4 μl), 2.5 mM dNTPs (2 μl), forward primer (0.8 μl), 5 μM reverse primer (0.8 μl), FastPfu Polymerase (0.4 μl), and template DNA (10 ng). Cycling proceeded as follows: 3 min at 95 °C twenty-seven cycles(30 sat 95 °C, 30 sat 55 °C, 45 sat 72 °C); 10 min at 72 °C. After amplicons extraction, samples were purified and quantified using the AxyPrep DNA Gel Extraction Kit (Axygen Biosciences, CA, USA) and QuantiFluor$^{TM}$-ST (Promega, Madison, WI, USA), respectively. Purified amplicons were pooled in equimolar proportions and paired-end sequenced (2 × 250 bp) on the Illumina MiSeq platform with TruSeqTM DNA Sample Prep Kit (Illumina, San Diego, CA, USA).

## 16S rRNA gene sequencing analysis

Raw fastq files were demultiplexed, quality-filtered by Trimmomatic and merged by FLASH with the following criteria: (i) The reads were truncated at any site receiving an average quality score <20 over a 50 bp sliding window; (ii) Primers were exactly matched allowing 2 nucleotide mismatching, and reads containing ambiguous bases were removed; (iii) Sequences whose overlap longer than 10 bp were merged according to their

overlap sequence. Operational taxonomic units (OTUs) were clustered with 97% similarity cutoff (*Edgar, 2013*) using UPARSE (version 7.1; http://drive5.com/uparse/) and chimeric sequences were identified and removed using UCHIME. The taxonomy of each 16S rRNA gene sequence was assigned by QIIME (version 1.7; http://qiime.org/home_static/dataFiles.html) (*Caporaso et al., 2010*) using RDP Classifier algorithm (http://rdp.cme.msu.edu/) (*Wang et al., 2007*) against the Silva (SSU123) 16S rRNA database (*Quast et al., 2013*) using a confidence threshold of 70%. Alpha diversity at the OTU level (e.g., Ace, Chao, Shannon and Simpson index) were calculated in QIIME following previously described methods (*Jiang & Takacs-Vesbach, 2017*; *Lauber et al., 2009*; *Van Horn et al., 2016*).

## Statistical analysis

Results analysis and figure generation based on clinicopathological information, alpha estimators and relative bacterial abundance were performed using SPSS 21.0 and GraphPad Prism 5.01 software. Student's $t$-test and the Mann–Whitney $U$-test were performed, with $p < 0.05$ indicating a significant difference between groups. Rarefaction curves were generated based on the alpha diversity estimators. The unweighted UniFrac algorithm was applied for hierarchical clustering and principal coordinates analysis at the OTU level to analyze beta diversity. We applied "Vennerable" package in R software (version 3.3.3) for the generation of venn diagram based at the OTU level. The Circos software (http://circos.ca/software/download/circos/) was performed for the generation of collonearity diagram to visualize the corresponding abundance relationship between samples and bacterial communities at the phylum and genus levels. The enriched and significant bacteria in each group were identified by linear discriminant analysis (LDA) combined with effect-size measurements (LEfSe) (*Segata et al., 2011*), with $p < 0.05$. For the Kruskal–Wallis test, LDA values >2 were considered significant (*Szafranski et al., 2015*).

# RESULTS

## Illumina sequencing data characteristics

The clinicopathological information for each of the three groups included in the study is presented in Table 1. There were no significant differences in terms of age or gender, while BMD, $T$-score and $Z$-score differed significantly among groups. Illumina sequencing captured a total of 694,232 high-quality sequences, with an average of 38,568 sequences/sample. Detailed information on the sequence results obtained for each sample are presented in Table S1.

## Inverse correlation between the number of bacterial taxa and the value of BMD

Based on the sequencing data, the gut microbiota of all samples were classified to 507 OTUs, 367 species, 235 genera, 99 families, 63 orders, 38 classes, 25 phyla. The number of bacterial taxa tended to increase at each level in accordance with the reduction in BMD, as shown in Table 2 and Fig. S1. Figure 1 presents a Venn diagram for the OP, ON and NC groups (at the OTU level). There were 455, 378, and 282 OTUs present in the OP, ON, and NC group, respectively. In addition, 208 OTUs (41%) were shared by all samples; 154 OTUs (30.4%)

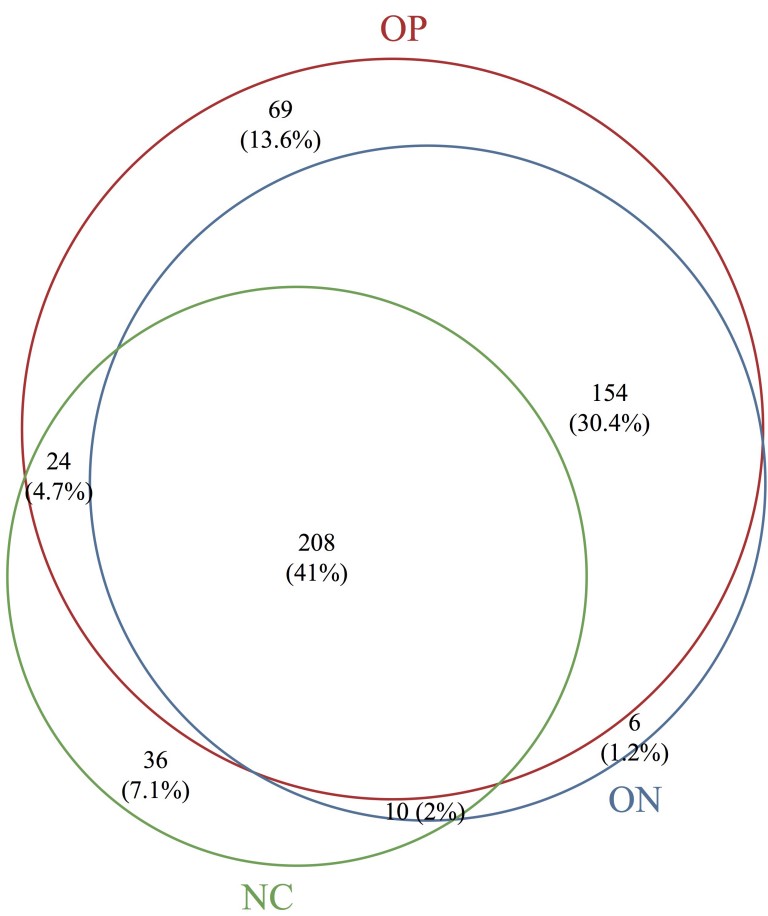

**Figure 1  Venn diagram of OP, ON and NC groups at OTU level.**

**Table 2  Bacterial taxa in each group at different levels.**

|       | Phylum | Class | Order | Family | Genus | Species | OTU |
|-------|--------|-------|-------|--------|-------|---------|-----|
| NC    | 8      | 14    | 20    | 41     | 134   | 218     | 282 |
| ON    | 21     | 33    | 56    | 88     | 195   | 296     | 378 |
| OP    | 23     | 35    | 58    | 92     | 219   | 335     | 455 |
| Total | 25     | 38    | 63    | 99     | 235   | 367     | 507 |

were shared between the OP and ON groups. For the remaining components (28.6%), the OP group (13.6%) accounted for nearly half of all OTUs.

## Diversity analysis of gut microbiota in osteoporosis and osteopenia patients

To determine alpha diversity, we calculated the mean ace index, chao index, shannon index, and simpson reciprocal index. This process allowed us to fully characterize the bacterial community diversity in samples. Detailed information on the estimators in each sample is presented in Table S2. The OTU level rarefaction curves of diversity estimators reached plateau phase (Fig. S2), indicating that most bacterial species had been captured

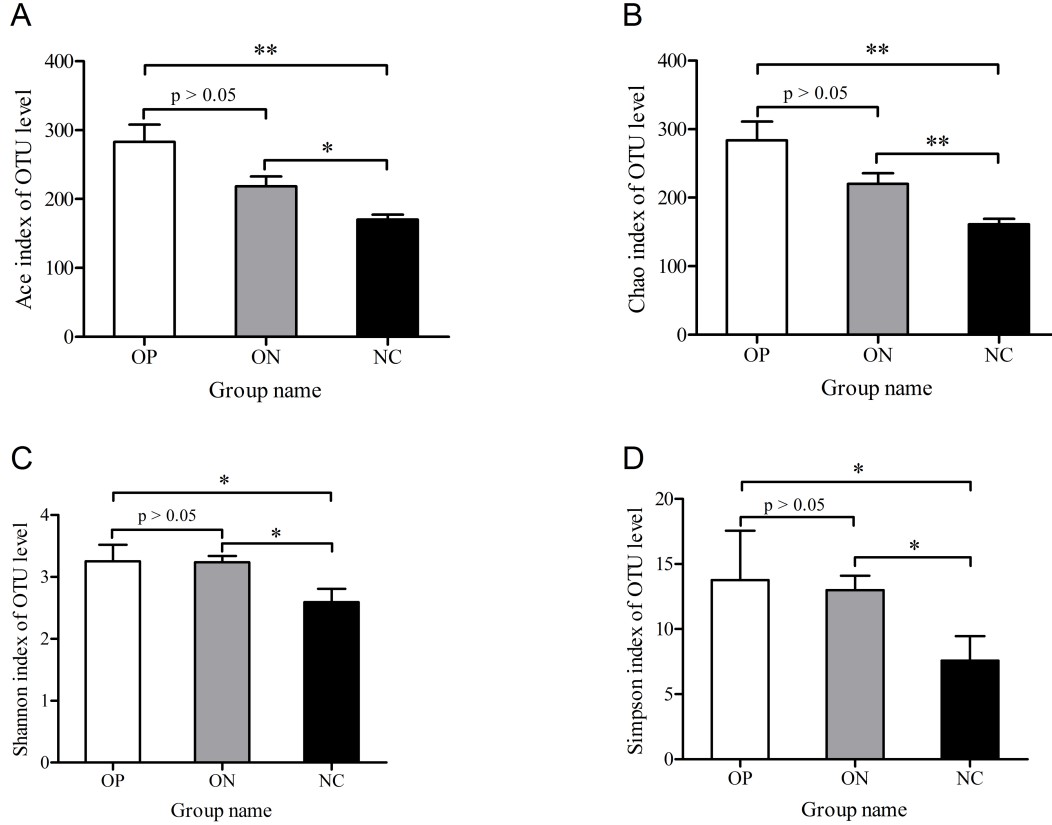

**Figure 2** **Significance of alpha diversity estimators between different groups.** *$0.01 < p \leq 0.05$, **$0.001 < p \leq 0.01$.

by sequencing in all samples. Higher numbers of the estimators represent greater diversity, which suggests that alpha diversity index was inversely correlated with BMD, although there were no significant differences between the OP and ON groups, as shown in Fig. 2.

With regard to beta diversity, unweighted UniFrac analysis indicated that hierarchical clustering and principal coordinate analysis (PCoA) could discriminate the NC samples from OP as well as ON samples. However, there was substantial overlap between the OP and ON groups, and most ON samples were positioned in the middle of the OP and NC samples, as Fig. 3 illustrates.

## Significance analysis of gut bacterial community abundance in osteoporosis and osteopenia patients

At the phylum level illustrated in Fig. 4, Firmicutes, Bacteroidetes, Proteobacteria and Actinobacteria constituted the four dominant phyla in all samples. The average ratios of Firmicutes/Bacteroidetes were 3.326, 1.755 and 1.290 in the OP, ON, and NC groups, respectively. Furthermore, we calculated the significance of the 10 most dominant phyla of microbial community structure among the OP, ON, and NC groups. Differences among the four dominant phyla (Firmicutes, Bacteroidetes, Proteobacteria, and Actinobacteria) were not statistically significant for comparisons between the OP and ON group or the ON and NC group ($p > 0.05$). Proportion of Firmicutes was significantly higher and Bacteroidetes

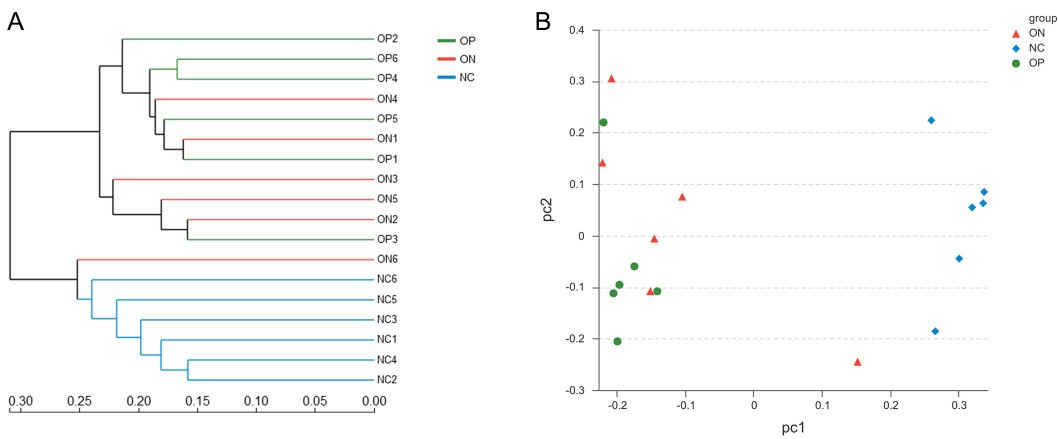

**Figure 3  Beta diversity analysis of OP, ON and NC group at OTU level.** (A) the hierarchical clustering tree. (B) Principal coordinate analysis (PCoA) scatter plot.

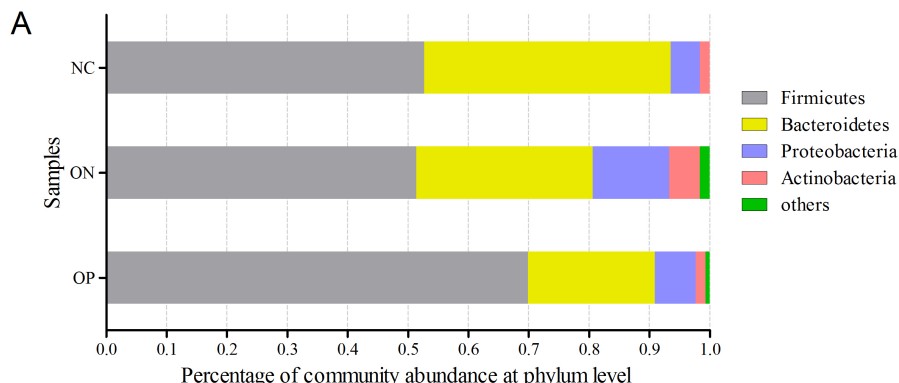

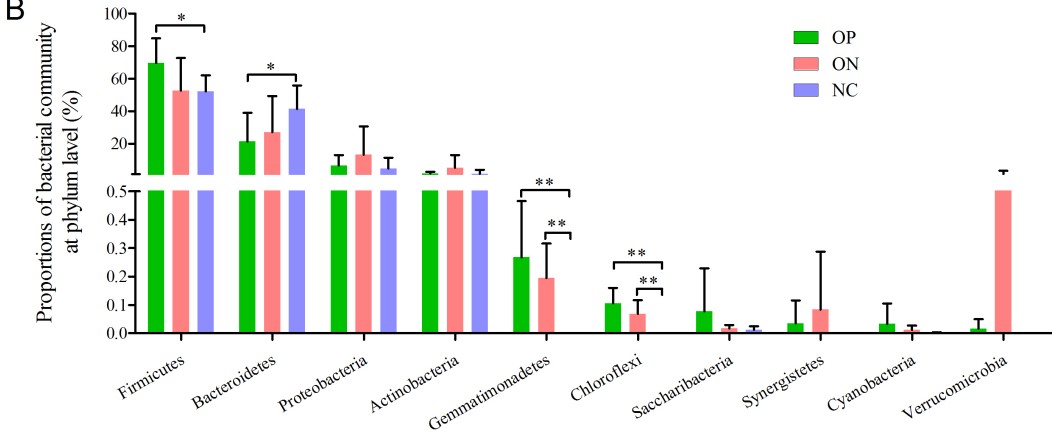

**Figure 4  Bacterial community abundance at phylum level of each group.** (A) Bacterial community abundance barplot at phylum level. (B) Significance of the top 10 bacterial community abundance at phylum level. *$0.01 < p \leq 0.05$, **$0.001 < p \leq 0.01$ based on Mann–Whitney $U$-test.

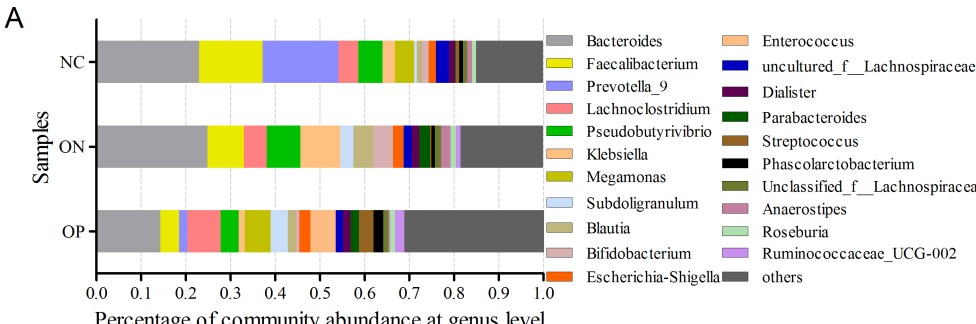

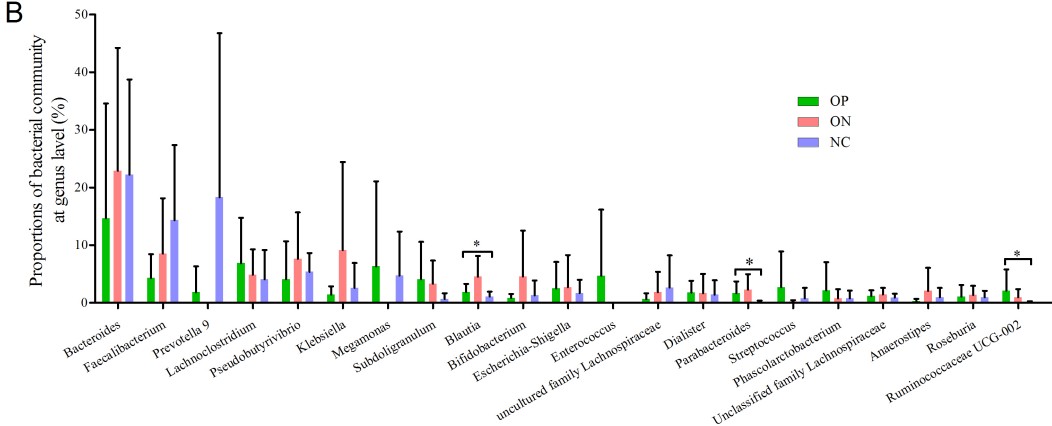

**Figure 5  Bacterial community abundance at genus level of each group.** (A) Bacterial community abundance barplot at genus level. (B) Significance of the 10 bacterial community abundance at genus level. *$0.01 < p \leq 0.05$ based on Mann–Whitney $U$-test.

proportion was significantly lower in OP samples than that in the NC group ($p < 0.05$) (Fig. 4B). As for other bacterial communities with small proportions, most of them were rare in the NC group but increased in the OP and ON groups. Gemmatimonadetes and Chloroflexi were significantly different between the OP and NC groups ($p < 0.01$) as well as between the ON and NC groups ($p < 0.01$).

At the genus level, a total of 21 genera with proportions above 1% were detected, as visualized in Fig. 5. *Bacteroides* accounted for the largest proportion in all samples. In the NC group, three genera (*Bacteroides*, *Faecalibacterium* and *Prevotella*) contributed more than half of the bacterial community. In the ON and OP groups, five and 11 genera, respectively, accounted for 50% of the bacterial community. Differentiation analysis of the 21 genera is presented in Fig. 5B. The *Blautia*, *Parabacteroides* and *Ruminococcaceae* genera differed significantly between the OP and NC groups. Fig. S3 depicts the collinearity diagram for the bacterial community and samples from all three groups.

We further applied linear discriminant analysis (LDA) combined effect size measurements (LEfSe) to explore the significant changes and relative richness of the bacterial community in the OP, ON, and NC groups, at phylum and genus levels. Fig. 6 summarizes the enrichment and variations in bacterial community for all three groups. At the phylum level, one phylum and seven phylum communities were enriched in the ON and

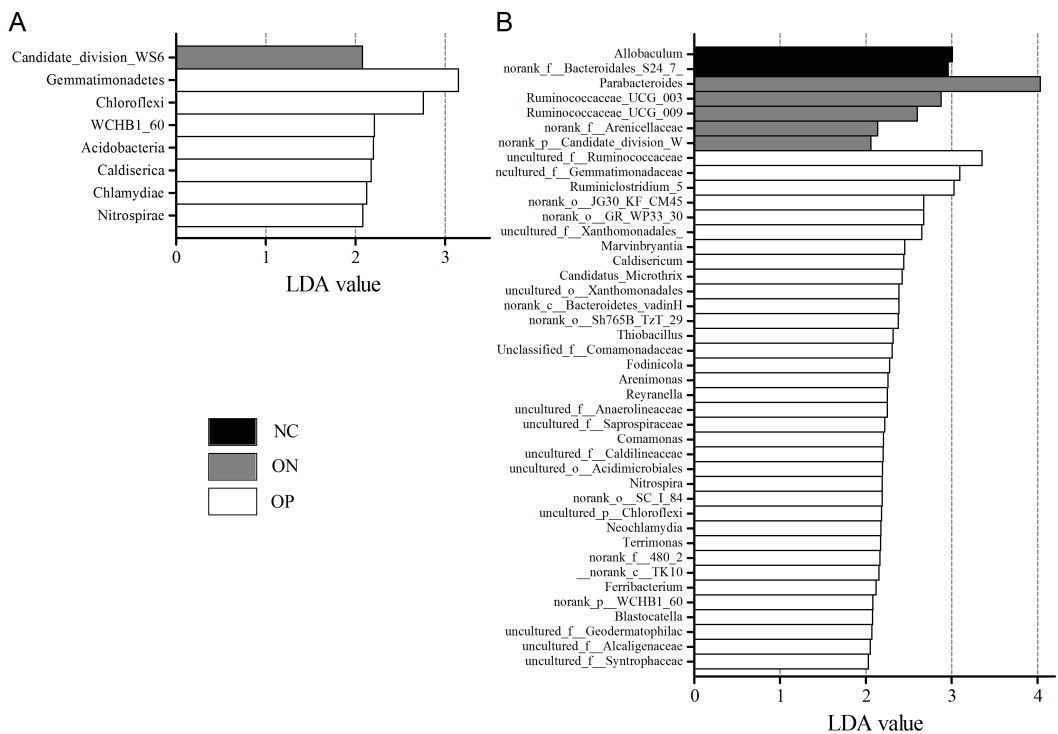

**Figure 6** **LEfSe at the phylum and genus level of each group.** (A) LEfSe bar at phylum level. (B) LEfSe bar at genus level. $P < 0.05$, LDA value > 2.

OP group, respectively, while no community in the NC group was enriched. At the genus level, 35 genus, five genus and two genus communities were enriched in the OP, ON and NC groups, respectively. The significance and variance of bacterial communities, as determined by sequencing analysis, may help discriminate OP or ON patients from NC subjects.

## DISCUSSION

The human microbiome, referred to as our second genome, can influence genetic diversity, immunity and metabolism (*Grice & Segre, 2012*; *Solt, Kim & Offer, 2011*). All of the bacteria in specific samples can now be detected based on microbiota DNA sequencing. Research focused on gut microbiota and bone metabolism has recently emerged. Our study is the first survey about composition and diversity analysis of gut microbiota in osteoporosis, osteopenia patients and healthy controls using metagenomic sequencing. The results indicate that bacterial component structure and diversity are altered in osteoporosis and osteopenia patients as compared with normal controls; this supported the perspective that the bone health can be affected by the gut microbiota.

Microbiota diversity analysis is valuable for quantifying the bacterial component and relative richness of a specific community. Our investigation of alpha diversity revealed an elevation of diversity estimators in the OP and ON groups. Hierarchical clustering and PCoA analysis of beta diversity was able to discriminate the NC group from the OP and ON groups. These results suggested that a rich diversity of gut microbiota may be related to the

reduction of bone mass. In the OP group, the proportion of Firmicutes phyla increased and the proportion of Bacteroidetes decreased significantly ($p < 0.05$) compared with that in the NC group. Several communities present at low levels in the OP and ON groups were absent in the NC group (e.g., Gemmatimonadetes Chloroflexi and Synergistetes). At the genus level, 21 genera with proportions over 1% were identified. *Bacteroides*, *Faecalibacterium* and *Prevotella* were the top 3 genera in the NC group, while *Prevotella* was not observed in the ON and was present at low levels in the OP group. The *Lachnoclostridium* and *Klebsiella* genera were more abundant in the OP and ON groups as compared to the NC group. We further identified the enriched and significant community in each group and speculated that these communities may be considered as specific biomarkers for the reduction of bone mass.

The underlying mechanisms of gut microbiota changes in osteoporosis and osteopenia patients remained to be explained. We hypothesized that the immune-inflammatory axis may act as the key bridge joining the gut microbiota to bone metabolism. Studies have shown that bone mass increased in germ-free (GF) mice compared with conventionally raised mice. The authors reported fewer osteoclasts, osteoclast precursor cells, CD4 (+) cells and inflammatory cytokines in the bone and bone marrow of GF mice. They also reported that bone mass could be normalized after gut microbiota transplantation in GF mice (*Sjogren et al., 2012*). Moreover, certain pre- and probiotics have been shown to increase bone mass ((*Bindels et al., 2015*; *Maekawa & Hajishengallis, 2014*; *Scholz-Ahrens et al., 2007*). Research suggests that gut microbiota and specific probiotics may regulate IGF-1, TNF-α and IL-1β, resulting in changes in bone formation and growth (*Ohlsson et al., 2014*; *Yan et al., 2016*).

Notably, this study does have certain limitations. The sample size may not have been large enough. The average age in the osteoporosis and osteopenia groups was 70 years, and the sex ratio of female:male is 5:1 in the two groups. The occurrence of osteoporosis is more common with age, and is more common in females than males. It was reported that the osteoporosis prevalence ranged from 9% to 38% for females and 1% to 8% for males in different countries (*Wade et al., 2014*). In this study, the subjects in the OP and ON groups were chosen randomly according with the recruiting criteria, and we further recruited the normal controls also at the same age and sex ratio to keep a balance. In view of this, we should consider the relevant hormonal changes, with corresponding effects on bone metabolism, because postmenopausal women are at high risk for osteoporosis (*Cappola & Shoback, 2016*). Researchers have reported that prebiotics improve calcium absorption, calcium accretion in bone and BMD in adolescents as well as postmenopausal female subjects (*Roberfroid et al., 2010*). Thus, dietary intake (e.g., pre- or probiotics) may alter bone metabolism in both pre- and post-menopausal women.

According to recent reports, studies in microbiota research have increased, which focusing on exploring new approaches for disease diagnosis and treatment (*Castro-Nallar et al., 2015*; *Vernocchi, Chierico & Putignani, 2016*). In our research, we explored gut microbiota diversity changes in primary osteoporosis and osteopenia patients. Further studies are required to understand the gut microbiota as a regulator for bone mass and evaluate it as a novel biomarker for osteoporosis.

## ACKNOWLEDGEMENTS

We would like to thank Kuan Liu, Sales Engineer from Majorbio, Shanghai, for the technology guidance.

### Funding

This project was funded by the China Postdoctoral Science Foundation (No. 2017M613176, No. 2017M613177), the National Natural Science Foundation of China (No. 81601898) and the Research Foundation of Xi'an Hong-Hui Hospital (No. YJ2016013). The funders had no role in study design, data collection and analysis, decision to publish, or preparation of the manuscript.

### Grant Disclosures

The following grant information was disclosed by the authors:
China Postdoctoral Science Foundation: 2017M613176, 2017M613177.
National Natural Science Foundation of China: 81601898.
Research Foundation of Xi'an Hong-Hui Hospital: YJ2016013.

### Competing Interests

The authors declare there are no competing interests.

### Author Contributions

- Jihan Wang and Yangyang Wang performed the experiments, wrote the paper, prepared figures and/or tables.
- Wenjie Gao and Biao Wang contributed reagents/materials/analysis tools.
- Heping Zhao and Yuhong Zeng analyzed the data.
- Yanhong Ji and Dingjun Hao conceived and designed the experiments, reviewed drafts of the paper.

### Human Ethics

The following information was supplied relating to ethical approvals (i.e., approving body and any reference numbers):

Hong Hui Hospital, Xi'an Jiaotong University, Biomedical research ethics committee.

### Data Availability

The raw sequence data is available at GenBank with accession number SRP095870 at https://trace.ncbi.nlm.nih.gov/Traces/sra/?study=SRP095870.

### Supplemental Information

Supplemental information for this article can be found online at http://dx.doi.org/10.7717/peerj.3450#supplemental-information.

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
