# Peer review of "Diversity analysis of gut microbiota in osteoporosis and osteopenia patients"

_PeerJ, doi:10.7717/peerj.3450_

## Round 0.1 · original submission · Minor Revisions

Dear Dr. Wang,

Based on the comments of the reviewers, please give a careful look at the grammar and the English language in the entire manuscript. I appreciate you addressing the reviewer's concerns systematically. please pay special attention to reviewer 2, who has raised significant concerns about the manuscript.

Thank you

·

Basic reporting

This is a clear and interesting paper. Literature references are correct and adequate. The manuscript is well written, correctly organized and the length is adequate.
The results are interesting although as the authors pointed out, maybe the population was too small(?)

Experimental design

The experimental design, development of experiments, technologic al tools and ethics seems to be good enough

Validity of the findings

The manuscript satisfies all the requirements of this section although I do not feel able to analyze statistics. A referee expert in this field seems necessary to ensure the paper is robust.

Additional comments

I've read with attention the manuscript.
This is a clear and interesting paper. Literature references are correct and adequate. The manuscript is well written, correctly organized and the length is adequate.
The results are interesting although as the authors pointed out, maybe the population was too small(?).
The experimental design, development of experiments, technological tools and ethics seems to be good enough, however as I do not feel able to analyze statistics, another reviewer expert in this field seems necessary to ensure the paper is robust.enough.

L 135 and 138 Replace taxonomies by taxa

Reviewer 2 ·

Basic reporting

General comments:
This paper focuses on identifying gut microbiota difference in osteoporosis and osteopenia patients using MiSeq16S rRNA gene sequencing. The current manuscript mainly focuses on reporting the taxonomic difference among experimental groups. Ideally, authors should also discuss the roles of these microbes in human guts (i.e., what they are doing.) and why the changes in microbiota can lead to diseases. So, readers will have a whole picture about the influence of gut microbiota on human health.
The current manuscript has quite a few issues that detract from its publishing. Although some of new results are important, they are eclipsed by grammatical errors, a generally poor description and justification of experimental and methodological approaches. Presentation of results is also unclear and misleading. An in-depth discussion related to results is also needed.
The English writing should be improved significantly, because the grammatical errors are almost everywhere (I only list grammatical major errors in this review). I strongly suggest that authors invite a native English speaker to proofread their manuscript.
Based on the current manuscript quality, this manuscript does not provide sufficient evidence to warrant publication.

Experimental design

Specific Comments:
Abstract:
Line 14: “evidence” is an uncountable noun.
Line 18: Multiple definite articles are missing. (e.g., “the” number of bacterial taxonomy, “the” value of bone mineral density).
Line 20-21: Both clustering and PCoA are beta diversity analyses. I suggest revising this sentence like this -- “Beta diversity analyses based on hierarchical clustering and principal coordinate analysis (PCoA) …….”
Line 23: There is a link verb missing in between “Bacteroidetes” and “significantly lower”
Line 29-30: “1 and 7 phylum” and “35, 5 and 2 genus” are confusing. Authors should specify what these numbers (e.g., 1 and 7) are.
Introduction:
Generally, the goal of the study needs to be clarified. There is a lot of background information about the relationship between gut microbiota and diseases, but the background information needs to be connected to this study. The authors did not mention the goals or hypotheses of their research, which can confuse readers.
Line 41: “the elderly”. Consider using “the elder”
Line 46: “characteristic” should be plural.
Line 51: This sentence needs to be rephrased. “a rich ecosystem” is a misleading description. It is unclear whether the number of microbes is high or the diversity of microbes is high.
Line 56-57: “Several reports have identified”. Consider revising it. For example, “Several studies have reported”.
Line 58-60: This sentence needs to be rephrased.
Line 60: There is a word “been” missing in between “has” and “performed”
Line 61-62: This sentence needs to be rephrased. It is unclear about the idea that authors wanted to express.
Line 67: Considering using “isolated” instead of “recovered”
Line 71-73: The last paragraph of introduction should be rewritten. Generally, the last paragraph of introduction should give a clear statement about the goal or hypothesis of current study. In this introduction, the authors only listed the experiment design and the approach.
Methods:
Generally, more information about the DNA extraction, sequencing and data analyses is needed. I suggest the author checking the “Methods” sections in the following papers or cite the following papers (These papers were also used software QIIME to do the analyses)
Jiang, X., and Takacs-Vesbach, C.D. (2017) Microbial community analysis of pH 4 thermal springs in Yellowstone National Park. Extremophiles 21: 135-152.
Van Horn, D.J., Wolf, C.R., Colman, D.R., Jiang, X., Kohler, T.J., McKnight, D.M. et al. (2016) Patterns of bacterial biodiversity in the glacial meltwater streams of the McMurdo Dry Valleys, Antarctica. FEMS Microbiol Ecol 92.
Furthermore, the experimental design can potentially lead to bias results, because of the uneven sex ratio in each group (F:M = 5: 1).
Line 100: Consider revising “27x” to twenty-seven cycles.
Line 103-104: Detailed sequencing information is missing here. The authors need to mention the sequencing reagent kits that they used.
Line 104: “2X250” should be “2X250bp”
Line 107: The version of QIIME seems to be incorrect. Does it mean version 1.7?
Line 107: There is a lot of information missing here. The authors should mention how they did the quality control for the raw reads. For example, did they join the paired-end reads? How did they check chimaeras?
Line 109: “aligned to SILVA database”. It is unclear whether SILVA database was also used for the taxonomic assignment. The database used for taxonomic assignment should be also given here.
Line 109: “database(Quast et al. 2013)”. Missing a space.
Line 111-115: Here lacks of detailed information about alpha and beta diversity analyses. Normally, alpha (e.g., Chao1, ACE, Shannon, etc.) and beta diversity (e.g., UniFrac) analyses require dataset normalization. This can be easily done by rarefying to equal depth (i.e., equal number of sequences across samples). Please check the above-mentioned two papers for details.
Line 113: “unweighted_unifrac”. Revise it to “unweighted UniFrac”
Line 114: “principal co-ordinates analysis”. I suggest that the authors write the terminologies consistently. They used “principal coordinates analysis” in the abstract (Line 20, without a “hyphen”).
Line 116: “The collinearity diagram”. There is no information about which software was used to build this diagram. Also, the Venn diagram were built (Figure 1). Currently, QIIME software does not have the functions to build these two types of diagrams. If authors used the software other than QIIME to build the diagrams, the name of software should be mentioned.
Line 118: “(LEfSe)”. A citation about this software is missing here.

Validity of the findings

Results:
Line 132: “with an average of 38568.44 sequences/sample”. The count of sequences is normally given as an integer.
Line 137: “1 kingdom”. Consider removing it. The term of Kingdom is rarely used in microbial ecology studies. Since you used Bacterial specific primers, you could only get the sequences from the Bacteria domain.
Line 140: “OTU level”. Again, authors need to keep the writing style consistently. For example, “OTU-level” was used in Line 149.
Line 152-153: “although there were no significant differences between the OP and ON groups, as shown in Figure 2.” In Figure 2, ACE and Chao indices suggest all three groups are significantly different (asterisk symbols on all three groups). This should be clarified.
Line 158-160: “Results of the diversity analysis suggest that a study of the gut microbiota may help researchers to understand osteoporosis and osteopenia diseases, which result from abnormal bone metabolism.” This does not belong to the “results” and this sentence should be in the “discussion” section.
Line 173: “most were almost non-existent”. This needs to be replaced by a more precise description.
Line 177. “1% were captured”. Consider revising it to “1% were identified/detected”
Line 189. “1 and 7 phylum” should be clarified.
Line 191. “35, 5 and 2 genus” should be clarified.

Discussion:
Generally, the discussion should be reorganized and rewritten. The discussion should be focused more on linking the changes in microbial communities to the diseases. The currently discussion merely repeated the results section. For example, they found that the Firmicutes in OP group were significantly higher compared to NC group, but they did not discuss why people have high proportion of Firmicutes will get OP disease. What is the function/role of the Firmicutes? Also, authors listed several genera that were found in different groups, but they did not specify if they are important or not. For example, OP and ON patients may be lack of a group of bacteria compared to normal people, which can be the cause of these diseases. Authors should also discuss why these bacteria are so important for human to stay healthy.
Line 199. “among the first surveys”. This sentence needs to be rephrased. It should be only one “first survey”.
Line 212-213. The text format here is incorrect.
Line 213-218. Basically, these sentences belong to results. Further discussion related to these results is needed.
Line 241. The authors should also mention the limitation of the uneven sex ratio in each group (F:M = 5:1).

Tables and Figures:
Figure 2. More information about statistic results is needed. For example, the top two figures can be misleading. It is unclear whether significant differences were found between OP and NC or significant differences were found among OP, ON and NC.
Figure 3B and 3D. The authors did not mention these figures in results or discussion. If these are not important figures, please consider removing them or showed them in the supplemental materials.
Figure 3A, 3B and 3C. It would be better to use “at OTU level” than “on OTU level”.
Figure 4B. The Y-axis of this figure needs to be clarified. It seems that there are two different scales on Y-axis.
Figure 5A. Some of genus names should be edited. For example, “Unclassified_f__Lachnospiracea” should be “Unclassified family Lachnospiracea”. All underlines within a name needs to be removed.
Figure 4 and 5. “*0.01<p≤0.05”. I suggest that authors include the name of the statistic method. For example, you can write like this -- “*0.01<p≤0.05 based on the ANOVA test”.
Figure S2. Consider using names as “Ace rarefication curves”, “Chao rarefication curves”, “Shannon rarefication curves” and “Simpson rarefication curves”.
Table 2. Consider removing the “Kingdom” column.
Table S1. Consider rounding off decimals for the “Mean_length”.

Additional comments

In general, the paper could use more editing and care from the authors. Authors should carefully check grammar (e.g., single/plural, countable/uncountable nouns, the/a, passive voice etc.).

·

Basic reporting

English language can be given a relook

Experimental design

Okay, but certain comments are included in the word file

Validity of the findings

May go through the word document

Additional comments

Authors need to consider the overall scenario in making an approach to the study. It should not be one fragment at one time.

·

Basic reporting

The manuscritpt entitled "Diversity analysis of gut microbiota in osteoporosis and osteopenia patients" is interesting. The autors showed a suggested relationship between the microbiota composition and the BMD. This manuscrit is well written and easy to follow.
References are correct and are in an appropiated number.

Experimental design

The experimental study are well done, and the methodology the develoment of the experiments seemed to be good enough.
The study presented the approval from a ethical comittee, and the authors obtained the informed consent from the participants.

One of the main issue of the manuscript is that the population of the study was low, as authors mentioned in the manuscript. As well as to study woman in pre and post menopausic situation.

Validity of the findings

The data obteined by the authors is quite robust, but probably needs more population in the study

Additional comments

Some suggestions to the authors.

1-Line 87 and 88, These sentences are criteria of inclusion of the patients to the study. They should be mentioned in the Subject recruitment section.
2-The groups should be more homogeneous regarding the age of patients.
3-In table S1, it is difficults to understand what is the meaning of sample named as total..
4-Figure 1 did not represented the percentatges found inside. Please choose another type of graph.
5-In figure 2, Are there statistical difference between OP an NC in Shanon and Simpson index? if yes, indicate it in the figure.
6-Line 131, please define T-score and Z-score.

---

## Round 0.2 · Minor Revisions

Dear Dr. Wang,

thank you for addressing the concerns of the reviewers and improving the manuscript significantly. I am happy to state that, I am ready to accept the manuscript with minor revisions. Please see below, the comments from Reviewer 2.

Reviewer 2 ·

Basic reporting

Dear Editor,
The manuscript has been significantly improved after the revision. However, the authors need to revise the following concerns, before this manuscript is officially published.
I noticed the authors deleted this sentence (“Alpha diversity at the OTU level (e.g., ace, chao, shannon and simpson index) were calculated.”) in their second manuscript. Alpha diversity is important to this paper and it should not be omitted. I strongly suggest the author add the following sentences and citations at the end of section “16S rRNA gene sequencing analysis” (Line 115).
I strongly suggest including the writing like this (add to the end of Line 115):
“Alpha diversity at the OTU level (e.g., Ace, Chao, Shannon and Simpson index) were calculated in QIIME following previously described methods (Jiang & Takacs-Vesbach 2017; Lauber et al. 2009; Van Horn et al. 2016)”

References
Jiang X, and Takacs-Vesbach CD. 2017. Microbial community analysis of pH 4 thermal springs in Yellowstone National Park. Extremophiles 21:135-152. 10.1007/s00792-016-0889-8
Lauber CL, Hamady M, Knight R, and Fierer N. 2009. Pyrosequencing-based assessment of soil pH as a predictor of soil bacterial community structure at the continental scale. Applied and Environmental Microbiology 75:5111-5120. 10.1128/aem.00335-09
Van Horn DJ, Wolf CR, Colman DR, Jiang X, Kohler TJ, McKnight DM, Stanish LF, Yazzie T, and Takacs-Vesbach CD. 2016. Patterns of bacterial biodiversity in the glacial meltwater streams of the McMurdo Dry Valleys, Antarctica. FEMS Microbiol Ecol 92. 10.1093/femsec/fiw148

Experimental design

Everything is good after revision.

Validity of the findings

Everything is good after revision.

Additional comments

Dear Editor,
The manuscript has been significantly improved after the revision. However, the authors need to revise the following concerns, before this manuscript is officially published.
I noticed the authors deleted this sentence (“Alpha diversity at the OTU level (e.g., ace, chao, shannon and simpson index) were calculated.”) in their second manuscript. Alpha diversity is important to this paper and it should not be omitted. I strongly suggest the author add the following sentences and citations at the end of section “16S rRNA gene sequencing analysis” (Line 115).
I strongly suggest including the writing like this (add to the end of Line 115):
“Alpha diversity at the OTU level (e.g., Ace, Chao, Shannon and Simpson index) were calculated in QIIME following previously described methods (Jiang & Takacs-Vesbach 2017; Lauber et al. 2009; Van Horn et al. 2016)”

References
Jiang X, and Takacs-Vesbach CD. 2017. Microbial community analysis of pH 4 thermal springs in Yellowstone National Park. Extremophiles 21:135-152. 10.1007/s00792-016-0889-8
Lauber CL, Hamady M, Knight R, and Fierer N. 2009. Pyrosequencing-based assessment of soil pH as a predictor of soil bacterial community structure at the continental scale. Applied and Environmental Microbiology 75:5111-5120. 10.1128/aem.00335-09
Van Horn DJ, Wolf CR, Colman DR, Jiang X, Kohler TJ, McKnight DM, Stanish LF, Yazzie T, and Takacs-Vesbach CD. 2016. Patterns of bacterial biodiversity in the glacial meltwater streams of the McMurdo Dry Valleys, Antarctica. FEMS Microbiol Ecol 92. 10.1093/femsec/fiw148

---

## Round 0.3 · accepted · Accept

Dear Dr. Wang,
Thank you for addressing the concerns and improving the manuscript.

Reviewer 2 ·

Basic reporting

The authors addressed all the reviewer concerns and I therefore recommend to publish this manuscript.

Experimental design

NA

Validity of the findings

NA

Additional comments

The authors addressed all the reviewer concerns and I therefore recommend to publish this manuscript.